# Enhanced Multi-Instance Partial Label Learning
# via Average Gradient Outer Product

**Nan Cao** [* 1]   **Xu Zhao** [* 1]   **Teng Zhang** [1]

## Abstract

Multi-Instance Partial-Label (MIPL) learning is a *dual* weakly supervised framework, where each training bag is annotated with a candidate label set comprising a single ground-truth label alongside multiple false positives. The primary challenge in MIPL is that the correspondence between key instances and the ground-truth label is significantly obscured by such coarse and ambiguous supervision. Existing methods largely inherit mechanisms from Multi-Instance Learning (MIL) and Partial-Label Learning (PLL) in isolation; they fail to address the interaction between these two forms of ambiguity, rendering them susceptible to false positives and often causing them to overlook key instances. In this paper, we leverage the Average Gradient Outer Product (AGOP) to adaptively rescale the feature space, thereby amplifying discriminative feature directions to facilitate the identification of key instances and ground-truth labels. By integrating AGOP with an attention module and a progressive disambiguation strategy, we propose AGOPMIPL. Extensive experiments on four MIPL benchmarks and a real-world CRC-MIPL dataset demonstrate that AGOPMIPL consistently outperforms five state-of-the-art baselines, achieving a relative gain of up to $25.9\%$ on CRC-MIPL-KMeansSeg.

## 1. Introduction

While supervised learning has achieved remarkable success, its reliance on exhaustive, high-quality annotations presents a severe bottleneck due to prohibitive labeling costs. This limitation has catalyzed the rise of weakly supervised learn-

ing (Zhou, 2018; Li et al., 2019; Nodet et al., 2021). Among these paradigms, Multi-Instance Learning (MIL) (Dietterich et al., 1997; Zhou et al., 2009) has emerged as a prominent framework utilizing coarse bag-level labels. However, standard MIL often proves insufficient for complex real-world scenarios where supervision is not only coarse but also ambiguous. Consider image annotation via crowdsourcing: an image (bag) containing multiple objects often yields a candidate label set comprising both the ground truth and false positives. This scenario introduces a dual weak supervision challenge: uncertainty regarding which instances are informative (instance space) and which label within the candidate set is correct (label space). To address this compounded challenge, Multi-Instance Partial-Label (MIPL) learning (Tang et al., 2023) has been proposed, aiming to jointly disambiguate the label space and localize key instances under such weak supervision.

Similar to MIL, existing MIPL methods generally fall into two categories. **Instance-level methods** propagate the candidate label set to individual instances for local disambiguation (Tang et al., 2024b). While this strategy effectively shifts the Partial-Label Learning (PLL) (Cour et al., 2011) to instance level, it often neglects global bag-level context. **Bag-level methods** aggregate instances into a unified representation to perform disambiguation at bag level (Tang et al., 2023; 2024a; Yang et al., 2024; 2025). The aggregation weights are typically derived from inter-instance correlations or distributional priors. Consequently, candidate labels supervise the bag representation only indirectly, failing to guide the model toward feature directions that distinguish the true label. Under noisy supervision, the attention mechanism is easily distracted by irrelevant instances, causing the key instance to be overlooked. To examine this limitation, we evaluate four attention-based MIPL methods on the FMNIST-MIPL dataset with different noise ratios $r$. Figure 1(a) reports the Top-1 hit rate of key-instance localization, and Figure 1(b) reports the average attention rank of key instances, where a lower rank indicates higher attention. The Top-1 hit rate decreases and the average rank worsens as label noise increases. These results show that existing attention-based MIPL methods have limited ability to localize key instances under candidate-set noise.

---

*\*Equal contribution   [1]School of Computer Science and Technology, Huazhong University of Science and Technology, Wuhan, China.   Correspondence to: Teng Zhang <tengzhang@hust.edu.cn>.*

*Proceedings of the $43^{rd}$ International Conference on Machine Learning*, Seoul, South Korea. PMLR 306, 2026. Copyright 2026 by the author(s).

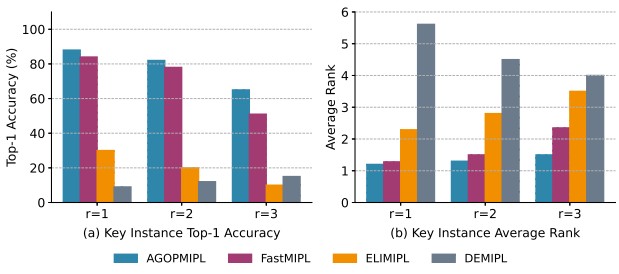

*Figure 1.* Key instance identification on FMNIST-MIPL with varying noise ratios $r$. (a) Top-1 hit rate. (b) Average rank under descending attention scores (lower is better).

Existing MIPL methods largely adapt mechanisms from MIL and PLL independently, failing to adequately address the interaction between these two forms of ambiguity. This limitation renders them susceptible to false positives and frequently leads to the neglect of key instances. In this paper, we propose AGOPMIPL, a novel framework based on the Average Gradient Outer Product (AGOP). Our approach is motivated by a pivotal observation: the gradients of the classifier with respect to the input contain rich information regarding discriminative feature directions, which remain robust even under label noise. Specifically, we utilize the average outer product of the classifier's Jacobian to explicitly identify these directions. By employing an AGOP-induced metric transformation, we rescale the feature space to amplify dimensions critical for classification. This transformation effectively suppresses noisy interference and steers the attention mechanism toward truly informative instances. Complementing this, we incorporate a prototype-based attention mechanism and a progressive label disambiguation module to ensure the robust identification of both key instances and ground-truth labels. Our contributions are fourfold:

- We propose an AGOP-based feature learning method that dynamically isolates discriminative features, enhancing the model's resilience to label ambiguity.
- We develop an attention mechanism that integrates AGOP-derived feature metrics with class prototypes, ensuring attention scores reflect both instance saliency and semantic alignment.
- We formulate a progressive disambiguation strategy that iteratively refines the label confidence distribution, enabling precise candidate label selection during training.
- We provide a theoretical analysis demonstrating that the AGOP transformation recovers the underlying discriminative subspace, theoretically guaranteeing the amplification of features aligned with the true label.

The remainder of this paper is organized as follows. Section 2 reviews related works. Section 3 details the proposed method. Section 4 presents a theoretical analysis of AGOP's effectiveness in MIPL. Section 5 reports empirical results, and Section 6 concludes the paper.

## 2. Preliminaries

### 2.1. Multi-Instance Learning

Multi-Instance Learning (MIL) (Dietterich et al., 1997) deals with the problem of learning from bags of instances using only bag-level supervision. Existing literature is broadly categorized into two streams: instance-level methods and bag-level methods (Amores, 2013; Carbonneau et al., 2018).

**Instance-level methods** focus on learning a classifier to label individual instances, which is subsequently coupled with a multi-instance assumption to infer bag labels. Numerous strategies originally developed for supervised learning have been adapted to construct such instance classifiers. For instance, Diverse Density (DD) (Maron & Lozano-Pérez, 1998) and its variant EM-DD (Zhang & Goldman, 2001) employ maximum likelihood estimation. ID3-MI (Chevaleyre & Zucker, 2001) adapts the standard ID3 decision tree, while MI-SVM (Andrews et al., 2002) and MI-ODM (Zhang & Jin, 2020) optimize the minimum margin and margin distribution, respectively. Furthermore, neural networks have been integrated into this framework, including FNNs (e.g., BP-MIP (Zhou & Zhang, 2002)), CNNs (e.g., EM-CNN (Hou et al., 2016)), and RNNs (e.g., SNL (Garcez & Zaverucha, 2012)).

**Bag-level methods** predict bag labels by treating each bag as a holistic entity, thereby bypassing the explicit inference of latent instance labels. One approach defines bag-wise distance or similarity functions, enabling the application of nearest neighbor and kernel methods; Citation-KNN (Wang & Zucker, 2000) and MI Kernels (Gärtner et al., 2002) are representative examples. A second approach maps each bag to a unified feature representation, facilitating the use of standard single-instance learning algorithms. This mapping is typically unsupervised, achieved via statistical aggregation (e.g., mean, min, max, and moments (Amores, 2013)) or instance feature aggregation (Wei et al., 2017).

**Recent deep bag-level methods.** With the development of deep learning, recent MIL methods extend the bag-level route by replacing hand-crafted bag mappings with learnable aggregation modules. Attention-based MIL (Ilse et al., 2018) provides a permutation-invariant pooling framework that learns instance weights during bag representation learning. Subsequent methods improve this framework by refining the aggregation signal through loss-aware objectives or feature distillation (Shi et al., 2020; Zhang et al., 2022b), modeling dependencies among instances (Shao et al., 2021; Pal et al., 2022), or interpreting attention-based aggregation under uncertainty (Cui et al., 2023). These methods improve

how a bag representation is formed under reliable bag-level labels, but they do not address the additional candidate-label ambiguity in MIPL, where false-positive labels may mislead the supervision used for key-instance localization.

## 2.2. Multi-Instance Partial-Label Learning

Multi-instance partial-label learning (MIPL) (Tang et al., 2023) extends MIL by replacing the reliable bag label with a candidate label set that contains one ground-truth label and several false-positive labels. This setting couples two sources of ambiguity: key-instance localization requires knowing which candidate label is correct, whereas label disambiguation requires a reliable bag representation formed from key instances. Therefore, errors in instance selection and errors in label selection can reinforce each other.

**Instance-level methods** perform disambiguation after assigning the bag-level candidate set to individual instances. This strategy converts a MIPL bag into a collection of partial-label instances and then aggregates instance predictions to infer the bag label. MIPLGP (Tang et al., 2024b) is a representative method, which learns soft labels for instances under candidate-label supervision. Such methods enable fine-grained disambiguation, but may weaken the global bag context and expose irrelevant instances to the same candidate labels as key instances.

**Bag-level methods** first map each bag to a holistic representation and then perform candidate-label disambiguation at the bag level. Attention-based methods such as DEMIPL (Tang et al., 2023) and ELIMIPL (Tang et al., 2024a) learn bag representations with disambiguation attention and refine candidate labels through additional disambiguation regularization. Recent methods further improve bag modeling through instance-importance pooling (Yang et al., 2025) or probabilistic modeling of the bag-label relation (Yang et al., 2024). These methods preserve the global bag context, but the candidate-label information is mainly used during or after aggregation, rather than to guide label-discriminative instance selection before the bag representation is formed.

AGOPMIPL addresses this limitation by using classifier-derived feature directions before attention aggregation. Specifically, AGOP reshapes the feature metric toward directions that are more relevant to label discrimination, so the attention module can better localize instances associated with the true label under candidate-set noise.

## 2.3. Average Gradient Outer Product

For a multi-class predictor $\boldsymbol{f} : \mathbb{R}^d \mapsto \mathbb{R}^q$, the (transposed) Jacobian matrix $\mathbf{J}_{\boldsymbol{f}}(\boldsymbol{x}) \in \mathbb{R}^{d \times q}$, evaluated at an input feature vector $\boldsymbol{x}$, characterizes the local sensitivity of the prediction to input perturbations. Given data $\{\boldsymbol{x}_i\}_{i \in [m]}$, where

$[m] \triangleq \{1, 2, \ldots, m\}$, the AGOP is defined as the matrix $\mathbf{G} = \sum_{i \in [m]} \mathbf{J}_{\boldsymbol{f}}(\boldsymbol{x}_i) \mathbf{J}_{\boldsymbol{f}}(\boldsymbol{x}_i)^\top / m \in \mathbb{R}^{d \times d}$. Intuitively, the diagonal entries of $\mathbf{G}$ quantify the average squared sensitivity of the prediction to individual features, thereby identifying the most discriminative dimensions. Conversely, the off-diagonal entries capture feature co-importance by revealing correlations between feature directions.

Recent theoretical analyses (Radhakrishnan et al., 2024) establish AGOP as a unifying mechanism for feature learning across diverse neural architectures, including FNNs, CNNs, RNNs, and Transformers. Building on this, the Recursive Feature Machine (RFM) (Radhakrishnan et al., 2024) leverages AGOP for explicit feature learning. Specifically, given the spectral decomposition $\mathbf{G} = \mathbf{V} \text{diag}(\boldsymbol{\lambda}) \mathbf{V}^\top$, RFM transforms the feature space via the mapping $\boldsymbol{x} \mapsto \mathbf{G}^{1/2} \boldsymbol{x}$, where $\mathbf{G}^{1/2} \triangleq \mathbf{V} \text{diag}(\sqrt{\boldsymbol{\lambda}}) \mathbf{V}^\top$. This operation rescales the feature space along informative directions, effectively implementing a learned Mahalanobis metric. In other words, the AGOP transformation aligns Euclidean distances in the projected space with the learned metric (Radhakrishnan et al., 2025). Despite its demonstrated efficacy in fully supervised settings, the potential of AGOP in weakly supervised scenarios remains largely underexplored.

## 3. Method

Figure 2 illustrates the training pipeline of AGOPMIPL. The method is designed to address the coupling between key-instance localization and candidate-label disambiguation in MIPL. To provide label-aware reference points, we first construct class-specific prototypes from bags associated with each candidate label and embed them together with bag instances. AGOPMIPL then uses the classifier-induced AGOP metric to rescale both instance and prototype features before attention aggregation, so that attention is computed in a feature geometry more aligned with label-discriminative directions. The resulting bag representation is used for prediction, and the prediction is further masked by the candidate set to update the soft training target progressively.

### 3.1. Feature Embedding

Let $\mathcal{X} \subseteq \mathbb{R}^d$ be the instance space and $\mathcal{Y} = [q]$ be the label set. A MIPL dataset comprising $m$ bags is denoted as $\mathcal{D} = \{(\{\boldsymbol{x}_{i,1}, \ldots, \boldsymbol{x}_{i,n_i}\}, \mathcal{S}_i)\}_{i \in [m]}$, where $\boldsymbol{x}_{i,j} \in \mathbb{R}^d$ for $\forall j \in [n_i]$ represents the instance and $\mathcal{S}_i \subseteq \mathcal{Y}$ denotes the candidate label set containing the unknown ground-truth label. For feature embedding, we apply a mapping function $\psi : \mathbb{R}^d \mapsto \mathbb{R}^{d'}$ to all the instances, yielding the embedded representation $\mathbf{X}_i = [\psi(\boldsymbol{x}_{i,1}), \ldots, \psi(\boldsymbol{x}_{i,n_i})]^\top \in \mathbb{R}^{n_i \times d'}$.

To construct class-specific prototypes, we aggregate all instances from bags associated with a specific class $l \in [q]$ and partition them into $n_p$ clusters. The resulting centroids,

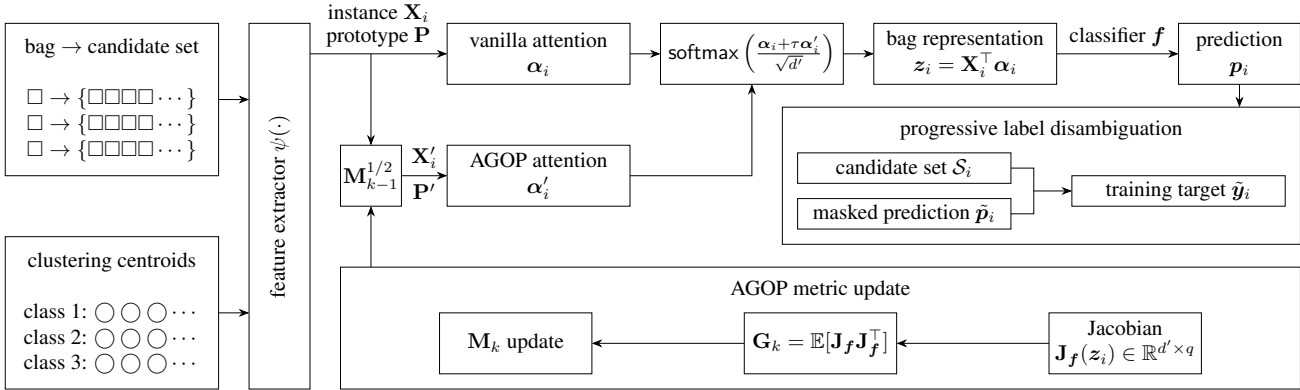

*Figure 2.* Overview of the AGOPMIPL framework. The feature extractor maps bag instances and class-specific prototypes into $\mathbf{X}_i$ and $\mathbf{P}$. The metric matrix $\mathbf{M}_{k-1}$ from the previous epoch transforms them into $\mathbf{X}'_i = \mathbf{X}_i \mathbf{M}_{k-1}^{1/2}$ and $\mathbf{P}' = \mathbf{P}\mathbf{M}_{k-1}^{1/2}$, enabling a dual-branch attention module to aggregate the bag representation $\mathbf{z}_i$. The classifier prediction is used for progressive label disambiguation, and the classifier Jacobian is used after the $k$-th epoch to compute $\mathbf{G}_k$ and update the metric matrix to $\mathbf{M}_k$.

$\mathbf{c}_{l,1}, \ldots, \mathbf{c}_{l,n_p}$, are mapped via $\psi$ and stacked to form the prototype matrix $\mathbf{P} = [\psi(\mathbf{c}_{1,1}), \psi(\mathbf{c}_{1,2}), \ldots, \psi(\mathbf{c}_{q,n_p})]^\top \in \mathbb{R}^{(qn_p) \times d'}$. In contrast to KISAR (Li et al., 2012), which clusters instances across all classes indiscriminately, our approach employs label-aware clustering to more effectively capture instance-label dependencies.

The architecture of $\psi$ is tailored to the data modality, typically comprising convolutional layers for spatial feature embedding and fully connected layers to project the outputs into a $d'$-dimensional space.

### 3.2. AGOP Transformation

Let $\boldsymbol{f}(\boldsymbol{z}) : \mathbb{R}^{d'} \mapsto \mathbb{R}^q$ denote the bag classifier. The (transposed) Jacobian matrix is denoted as $\mathbf{J}_{\boldsymbol{f}}(\boldsymbol{z}) \in \mathbb{R}^{d' \times q}$, with the $(j, l)$-th entry defined as $\partial f_l(\boldsymbol{z})/\partial z_j$. When $f$ is a deep neural network, the Jacobian can be computed via the chain rule using only forward propagation. Alternatively, if $f$ is easy to evaluate but expensive to differentiate, the Jacobian can be approximated via finite differences, which serves as a statistically consistent estimator. In our implementation, for simplicity, we employ a linear classifier; consequently, the Jacobian corresponds exactly to the classifier weights. This renders the Jacobian input-independent, allowing for efficient computation.

At each epoch $k$, we first transform the instances and prototypes using the metric matrix from the previous epoch: $\mathbf{X}'_i = \mathbf{X}_i \mathbf{M}_{k-1}^{1/2}$ and $\mathbf{P}' = \mathbf{P}\mathbf{M}_{k-1}^{1/2}$, where $\mathbf{M}_0$ is initialized as the identity matrix and $\mathbf{M}_{k-1}^{1/2}$ is computed analogously to $\mathbf{G}^{1/2}$ in Section 2.3. After the epoch, we compute the AGOP matrix $\mathbf{G}_k = \sum_{i \in [m]} \mathbf{J}_{\boldsymbol{f}}(\boldsymbol{z}_i) \mathbf{J}_{\boldsymbol{f}}(\boldsymbol{z}_i)^\top / m \in \mathbb{R}^{d' \times d'}$ using the current classifier. Then we update the metric matrix using Exponential Moving Average (EMA): $\mathbf{M}_k = \beta \mathbf{M}_{k-1} + (1 - \beta)\mathbf{G}_k$, where $\beta \in [0, 1)$ is the smoothing factor.

### 3.3. Feature Aggregation

To aggregate instance features into a comprehensive bag representation, we employ an attention mechanism that operates on both the original features $(\mathbf{X}_i, \mathbf{P})$ and the AGOP-transformed features $(\mathbf{X}'_i, \mathbf{P}')$.

Initially, we project the instances and prototypes into a shared latent space via $\mathbf{X}_i \mapsto \mathbf{X}_i \mathbf{W}_X$ and $\mathbf{P} \mapsto \mathbf{P}\boldsymbol{w}_P$, respectively, where $\mathbf{W}_X \in \mathbb{R}^{d' \times (qn_p)}$ and $\boldsymbol{w}_P \in \mathbb{R}^{d'}$ denote learnable parameters. Subsequently, we apply the sigmoid activation function $\sigma$ to the projected features and compute their Hadamard (element-wise) product, $\sigma(\mathbf{X}_i \mathbf{W}_X) \odot \sigma(\mathbf{P}\boldsymbol{w}_P) \in \mathbb{R}^{n_i \times (qn_p)}$. This operation utilizes broadcasting over the $n_i$ instances to effectively capture the alignment between instances and prototypes. Finally, we process this product through a linear transformation, a ReLU activation, and a subsequent linear mapping to yield the attention scores:

$$\boldsymbol{\alpha}_i = \mathsf{ReLU}([\sigma(\mathbf{X}_i \mathbf{W}_X) \odot \sigma(\mathbf{P}\boldsymbol{w}_P)]\mathbf{W})\boldsymbol{w} \in \mathbb{R}^{n_i},$$

where $\mathbf{W} \in \mathbb{R}^{(qn_p) \times D}$ and $\boldsymbol{w} \in \mathbb{R}^D$ are additional learnable parameters.

The attention scores $\boldsymbol{\alpha}'_i$ corresponding to the AGOP-transformed features $(\mathbf{X}'_i, \mathbf{P}')$ are computed analogously using a distinct set of parameters $\mathbf{W}'_X, \boldsymbol{w}'_P, \mathbf{W}', \boldsymbol{w}'$. The final attention scores are obtained by combining the both:

$$\boldsymbol{\alpha}_i \leftarrow \mathsf{softmax}((\boldsymbol{\alpha}_i + \tau\boldsymbol{\alpha}'_i)/\sqrt{d'}),$$

where $\tau$ is a scalar hyperparameter balancing the contribution of the two attentions. The final bag representation $\boldsymbol{z}$ is then derived by aggregating the instance features weighted by these attention scores: $\boldsymbol{z}_i = \mathbf{X}_i^\top \boldsymbol{\alpha}_i$.

## 3.4. Progressive Label Disambiguation

In MIPL learning, the ground-truth label is latent within a candidate set, where false positives can significantly mislead the training process. To mitigate this, we propose a progressive disambiguation strategy inspired by curriculum learning (Bengio et al., 2009). This approach initially relies on the provided candidate label set but gradually shifts reliance toward the model's own predictions as training proceeds. In the early stages, model predictions are inherently unreliable, necessitating dependence on the candidate set despite its noise. However, as the model learns discriminative patterns, its predictions become increasingly accurate, facilitating the identification of false positives. This progressive transfer of trust enables the model to bootstrap from noisy supervision toward accurate labels.

For the $i$-th bag, we define the candidate-label vector as $\boldsymbol{y}_i = \mathsf{norm}(\mathsf{mask}_{\mathcal{S}_i}(\mathbf{1}))$, where $\mathsf{mask}(\cdot)$ denotes the masking operation $\mathsf{mask}_\square(\boldsymbol{p}) = \boldsymbol{p} \odot [\mathbb{I}(1 \in \square), \dots, \mathbb{I}(q \in \square)]$*, and $\mathsf{norm}(\cdot)$ represents $\ell_1$-normalization, projecting the non-negative input onto the probability simplex. In the $t$-th iteration, let $\boldsymbol{p}_i^{(t)} = \mathsf{softmax}(\boldsymbol{f}(\boldsymbol{z}_i))$ denote the model's prediction, and $\tilde{\boldsymbol{p}}_i^{(t)} = \mathsf{norm}(\mathsf{mask}(\boldsymbol{p}_i^{(t)} + \epsilon\mathbf{1}))$ denote the masked prediction, where $\epsilon > 0$ is a small constant preventing zero probabilities. We formulate the training target as a convex combination of the candidate-label vector and the model's masked prediction: $\tilde{\boldsymbol{y}}_i^{(t)} = \rho_t \boldsymbol{y}_i + (1 - \rho_t)\tilde{\boldsymbol{p}}_i^{(t)}$. Within each AGOP epoch of $T$ iterations, the confidence factor $\rho_t$ follows a warmup-decay schedule. Specifically, $\rho_t = 1$ for the first $\lfloor \nu T \rfloor$ iterations where $\nu \in [0, 1)$ to prevent premature disambiguation. Subsequently, $\rho_t$ decays linearly as $\rho_t = (T - t)/(T - \nu T)$, reaching 0 at the epoch's end. This schedule resets at the beginning of each epoch as the AGOP-transformed features are updated.

## 3.5. Objective

The training objective comprises three distinct loss terms. First, the candidate mapping loss, defined as $\mathcal{L}_{\mathsf{mapping}} = -\sum_{i \in [m]} \sum_{c \in [q]} \tilde{y}_{i,c} \log(p_{i,c})$, aligns the model's predictions with the progressive training targets. Second, the sparsity loss, given by $\mathcal{L}_{\mathsf{sparsity}} = -\sum_{i \in [m]} \sum_{c \in [q]} \tilde{p}_{i,c} \log(\tilde{p}_{i,c})$, encourages the masked prediction $\tilde{\boldsymbol{p}}$ to concentrate probability mass on a single candidate label. Third, the non-candidate inhibition loss, formulated as $\mathcal{L}_{\mathsf{inhibition}} = -\sum_{i \in [m]} \sum_{c:c \notin \mathcal{S}_i} \log(1 - p_{i,c})$, penalizes high probabilities assigned to labels outside the candidate set. The final objective function is expressed as:

$$\mathcal{L} = \mathcal{L}_{\mathsf{mapping}} + \mu \mathcal{L}_{\mathsf{sparsity}} + \gamma \mathcal{L}_{\mathsf{inhibition}}, \qquad (1)$$

where $\mu, \gamma > 0$ are hyperparameters that balance the three loss terms.

---

*When the context is clear, we omit the explicit dependence of $\mathsf{mask}(\cdot)$ on $\mathcal{S}_i$.

Algorithm 1 presents the pseudo-code for the AGOPMIPL.

---

**Algorithm 1** AGOPMIPL

---

**Input:** Dataset $\mathcal{D} = \{(\mathbf{X}_i, \mathcal{S}_i)\}_{i \in [m]}$, number of clusters $n_p$, maximum epochs $K$, iterations per epoch $T$, EMA coefficient $\beta$

1: Initialize learnable parameters in the feature extractor $\psi$, attention modules, and classifier $\boldsymbol{f}$
2: $\forall i \in [m] : \boldsymbol{y}_i \leftarrow \mathsf{norm}(\mathsf{mask}_{\mathcal{S}_i}(\mathbf{1}))$ ▷ Candidate lab. vec.
3: $\forall l \in [q] :$ Perform clustering to obtain $\boldsymbol{c}_{l,1}, \dots, \boldsymbol{c}_{l,n_p}$
4: $\mathbf{M}_0 \leftarrow \mathbf{I}$ ▷ Metric matrix initialization
5: **for** $k = 1 \to K$ **do**
6:     **for** $t = 1 \to T$ **do**
7:        $\mathbf{P} \leftarrow [\psi(\boldsymbol{c}_{1,1}), \dots, \psi(\boldsymbol{c}_{q,n_p})]^\top$ ▷ Prototype
8:        $\mathbf{P}' \leftarrow \mathbf{P}\mathbf{M}_{k-1}^{1/2}$ ▷ AGOP transformation
9:        Compute $\rho_t$ via the warmup-decay schedule
10:       Sample mini-batch $\mathcal{I}_t \subseteq [m]$
11:       **for all** $i \in \mathcal{I}_t$ **do**
12:          $\mathbf{X}_i \leftarrow [\psi(\boldsymbol{x}_{i,1}), \dots, \psi(\boldsymbol{x}_{i,n_i})]^\top$ ▷ Instance
13:          $\mathbf{X}_i' \leftarrow \mathbf{X}_i\mathbf{M}_{k-1}^{1/2}$ ▷ AGOP transformation
14:          $\boldsymbol{\alpha}_i \leftarrow \mathsf{ReLU}([\sigma(\mathbf{X}_i\mathbf{W}_X) \odot \sigma(\mathbf{P}\boldsymbol{w}_P)]\mathbf{W})\boldsymbol{w}$
15:          $\boldsymbol{\alpha}_i' \leftarrow \mathsf{ReLU}([\sigma(\mathbf{X}_i'\mathbf{W}_X') \odot \sigma(\mathbf{P}'\boldsymbol{w}_P')]\mathbf{W}')\boldsymbol{w}'$
16:          $\boldsymbol{\alpha}_i \leftarrow \mathsf{softmax}((\boldsymbol{\alpha}_i + \tau\boldsymbol{\alpha}_i')/\sqrt{d'})$ ▷ Attention
17:          $\boldsymbol{z}_i \leftarrow \mathbf{X}_i^\top \boldsymbol{\alpha}_i$ ▷ Bag feature
18:          $\boldsymbol{p}_i \leftarrow \mathsf{softmax}(\boldsymbol{f}(\boldsymbol{z}_i))$ ▷ Prediction
19:          $\tilde{\boldsymbol{p}}_i \leftarrow \mathsf{norm}(\mathsf{mask}_{\mathcal{S}_i}(\boldsymbol{p}_i + \epsilon\mathbf{1}))$ ▷ Masked pre.
20:          $\tilde{\boldsymbol{y}}_i \leftarrow \rho_t \boldsymbol{y}_i + (1 - \rho_t)\tilde{\boldsymbol{p}}_i$ ▷ Progressive target
21:       **end for**
22:       Compute the total loss $\mathcal{L}$
23:       Update learnable parameters by one SGD step
24:     **end for**
25:     $\forall i \in [m] :$ Compute Jacobian $\mathbf{J}_{\boldsymbol{f}}(\boldsymbol{z}_i)$
26:     $\mathbf{G}_k \leftarrow \sum_{i \in [m]} \mathbf{J}_{\boldsymbol{f}}(\boldsymbol{z}_i)\mathbf{J}_{\boldsymbol{f}}(\boldsymbol{z}_i)^\top / m$ ▷ AGOP matrix
27:     **if** $\mathbf{G}_k$ has converged **then**
28:       **Break**
29:     **else**
30:       $\mathbf{M}_k \leftarrow \beta\mathbf{M}_{k-1} + (1 - \beta)\mathbf{G}_k$ ▷ Update metric matrix
31:     **end if**
32: **end for**

---

## 4. Analysis

In this section, we provide a theoretical justification for why AGOP can help MIPL without explicitly identifying the true label in the candidate set. The key observation is that true-label evidence and false-positive evidence behave differently across bags. Features associated with the true label tend to induce similar classifier sensitivity directions across bags of the same class, whereas false-positive labels usually depend on bag-specific irrelevant instances and therefore produce less coherent directions. AGOP exploits this dif-

ference by averaging Jacobian outer products over training bags: sensitivity directions that recur across bags accumulate, while inconsistent false-positive effects remain as a spectrally bounded background component. The resulting metric enlarges the stable discriminative subspace before attention aggregation, making key instances easier to separate under candidate-label noise.

Let $\boldsymbol{f} : \mathbb{R}^{d'} \mapsto \mathbb{R}^q$ denote the bag classifier. The Jacobian matrix $\mathbf{J}_{\boldsymbol{f}}$ admits a decomposition $\mathbf{J}_{\boldsymbol{f}} = \mathbf{J}_{\text{true}} + \mathbf{J}_{\text{false}}$, where $\mathbf{J}_{\text{true}}$ captures the sensitivity of the true label to the key instances, while $\mathbf{J}_{\text{false}}$ represents the sensitivity of false positives to irrelevant instances. We posit the following assumptions:

**Assumption 4.1.** $\mathbb{E}[\mathbf{J}_{\text{true}}\mathbf{J}_{\text{true}}^{\top}] = \mathbf{U}\boldsymbol{\Lambda}\mathbf{U}^{\top}$, where the expectation is taken with respect to the bag feature distribution, $\mathbf{U} = [\boldsymbol{u}_1, \ldots, \boldsymbol{u}_s] \in \mathbb{R}^{d' \times s}$ consists of orthogonal columns spanning an $s$-dimensional subspace of $\mathbb{R}^{d'}$, and $\boldsymbol{\Lambda} = \text{diag}(\lambda_1, \ldots, \lambda_s)$ with $\lambda_1 \geq \cdots \geq \lambda_s > 0$. This spectral decomposition implies that the sensitivity to the true label is concentrated primarily within the subspace $\text{Col}(\mathbf{U})$.

**Assumption 4.2.** Let $\boldsymbol{\Sigma}_{\text{false}} = \mathbb{E}[\mathbf{J}_{\text{false}}\mathbf{J}_{\text{false}}^{\top}]$ denote the second-moment matrix of the false-positive sensitivity. There exist constants $0 < \sigma_- \leq \sigma_+$ such that

$$\sigma_-^2\mathbf{I} \preceq \boldsymbol{\Sigma}_{\text{false}} \preceq \sigma_+^2\mathbf{I}.$$

This assumption does not require the false-positive directions to be orthogonal or perfectly isotropic. It only requires their energy to be spectrally bounded, so that candidate noise does not concentrate strongly along a few dominant directions.

**Assumption 4.3.** $\mathbb{E}[\mathbf{J}_{\text{true}}\mathbf{J}_{\text{false}}^{\top}] = \mathbf{0}$; that is, $\mathbf{J}_{\text{true}}$ and $\mathbf{J}_{\text{false}}$ are uncorrelated in expectation.

Based on these assumptions, we prove the theorem.

**Theorem 4.4.** *Let $\mathbf{V} = [\boldsymbol{v}_1, \ldots, \boldsymbol{v}_{d'-s}] \in \mathbb{R}^{d' \times (d'-s)}$ be a matrix such that $[\mathbf{U}, \mathbf{V}]$ forms an orthonormal matrix. For any vector $\boldsymbol{x} = \boldsymbol{x}_U + \boldsymbol{x}_V$ with $\boldsymbol{x}_U \in \text{Col}(\mathbf{U})$ and $\mathbf{0} \neq \boldsymbol{x}_V \in \text{Col}(\mathbf{V})$, the AGOP transformation $\boldsymbol{x} \mapsto \mathbf{G}^{1/2}\boldsymbol{x}$ satisfies:*

$$\frac{\|\mathbf{G}^{1/2}\boldsymbol{x}_U\|}{\|\mathbf{G}^{1/2}\boldsymbol{x}_V\|} \geq \sqrt{\frac{\lambda_s + \sigma_-^2}{\sigma_+^2}} \cdot \frac{\|\boldsymbol{x}_U\|}{\|\boldsymbol{x}_V\|}. \qquad (2)$$

*In particular, the discriminative component is amplified relative to the orthogonal component when $\lambda_s > \sigma_+^2 - \sigma_-^2$.*

*Proof.* By Assumptions 4.1–4.3, the AGOP matrix satisfies

$$\begin{aligned} \mathbf{G} &= \mathbb{E}[\mathbf{J}_{\boldsymbol{f}}\mathbf{J}_{\boldsymbol{f}}^{\top}] \\ &= \mathbb{E}[(\mathbf{J}_{\text{true}} + \mathbf{J}_{\text{false}})(\mathbf{J}_{\text{true}} + \mathbf{J}_{\text{false}})^{\top}] \\ &= \mathbb{E}[\mathbf{J}_{\text{true}}\mathbf{J}_{\text{true}}^{\top}] + \mathbb{E}[\mathbf{J}_{\text{true}}\mathbf{J}_{\text{false}}^{\top}] \end{aligned}$$

$$\begin{aligned} &\quad + \mathbb{E}[\mathbf{J}_{\text{false}}\mathbf{J}_{\text{true}}^{\top}] + \mathbb{E}[\mathbf{J}_{\text{false}}\mathbf{J}_{\text{false}}^{\top}] \\ &= \mathbf{U}\boldsymbol{\Lambda}\mathbf{U}^{\top} + \boldsymbol{\Sigma}_{\text{false}}. \end{aligned}$$

For $\boldsymbol{x}_U \in \text{Col}(\mathbf{U})$, we have

$$\begin{aligned} \|\mathbf{G}^{1/2}\boldsymbol{x}_U\|^2 &= \boldsymbol{x}_U^{\top}\mathbf{G}\boldsymbol{x}_U \\ &= \boldsymbol{x}_U^{\top}\mathbf{U}\boldsymbol{\Lambda}\mathbf{U}^{\top}\boldsymbol{x}_U + \boldsymbol{x}_U^{\top}\boldsymbol{\Sigma}_{\text{false}}\boldsymbol{x}_U \\ &\geq (\lambda_s + \sigma_-^2)\|\boldsymbol{x}_U\|^2. \end{aligned}$$

For $\boldsymbol{x}_V \in \text{Col}(\mathbf{V}) = \text{Col}(\mathbf{U})^{\perp}$, we have $\mathbf{U}^{\top}\boldsymbol{x}_V = \mathbf{0}$, and therefore

$$\begin{aligned} \|\mathbf{G}^{1/2}\boldsymbol{x}_V\|^2 &= \boldsymbol{x}_V^{\top}\mathbf{G}\boldsymbol{x}_V \\ &= \boldsymbol{x}_V^{\top}\boldsymbol{\Sigma}_{\text{false}}\boldsymbol{x}_V \\ &\leq \sigma_+^2\|\boldsymbol{x}_V\|^2. \end{aligned}$$

Taking the ratio of square roots yields Eqn. (2). □

Eqn. (2) formalizes this mechanism. AGOP does not remove false-positive labels directly; instead, it changes the geometry in which attention is computed. When the true-label sensitivity along $\text{Col}(\mathbf{U})$ is stronger than the spectral spread of the false-positive component, the AGOP transform stretches true-label-aligned directions more than orthogonal noise directions. Thus, key-instance features become more distinguishable before the model performs attention-based bag aggregation.

## 5. Experiments

To evaluate AGOPMIPL, we conduct comparative experiments against other MIPL methods on both benchmark and real-world datasets.[†] Furthermore, ablation studies are performed to assess the contribution of the AGOP module, and a sensitivity analysis of the hyperparameters is also carried out.

We use four benchmark MIPL datasets: MNIST-MIPL, FMNIST-MIPL, BIRDSONG-MIPL, and SIVAL-MIPL. We also evaluate on CRC-MIPL, a real-world colorectal cancer classification dataset containing 7,000 hematoxylin and eosin staining images from colorectal cancer and normal tissues. CRC-MIPL provides four bag constructions: Row, single blob with neighbors (SBN), k-means segmentation (KMeansSeg), and scale-invariant feature transform (SIFT). Its candidate label sets are provided by three crowdsourcing workers without expert pathologist annotations.

Table 1 summarizes the dataset statistics. #bags, #ins, #dim, #class, and avg. #CLs denote the number of bags, the number of instances, the instance feature dimension, the number

---

[†] Our implementation is available at https://github.com/LittleSpecial/AGOPMIPL.

of classes, and the average candidate-label set size, respectively. max #ins, min #ins, and avg. #ins denote the maximum, minimum, and average number of instances per bag.

We compare AGOPMIPL with five MIPL baselines: MI-PLGP (Tang et al., 2024b), DEMIPL (Tang et al., 2023), ELIMIPL (Tang et al., 2024a), ProMIPL (Yang et al., 2024), and FastMIPL (Yang et al., 2025). We also include two MIL methods, B-rFF (Pal et al., 2022) and Loss-ATTEN (Shi et al., 2020), and two PLL methods, PRODEN (Lv et al., 2020) and CAVL (Zhang et al., 2022a). Due to space limitations, the main paper reports the MIPL results, and the MIL and PLL results are provided in the supplementary material. We tune all baselines following the settings recommended in their original papers. For the benchmark datasets, the number of false-positive candidate labels is controlled by $r \in \{1, 2, 3\}$.

AGOPMIPL is implemented in PyTorch and trained on an NVIDIA A800 GPU. We use SGD with a momentum of 0.9 and a weight decay of 0.0001. For MNIST-MIPL and FMNIST-MIPL, the feature extractor consists of a two-layer convolutional network followed by a fully connected network. For Birdsong-MIPL and SIVAL-MIPL, we use a fully connected network on the preprocessed features. For CRC-MIPL, we use a fully connected network after each image bag generator or ResNet feature extractor.

The learning rate is selected from $\{0.0005, 0.001, 0.002, 0.005\}$. The maximum number of epochs is 200 for MNIST-MIPL and FMNIST-MIPL, and 400 for the other datasets. We split each dataset into training and test sets with a 7:3 ratio. We repeat the experiments over ten random train/test splits and report the mean accuracy and standard deviation. The best result is highlighted in bold.

**Noise injection protocol.** We make the candidate-noise level explicit. For each bag in the synthetic MIPL benchmarks, $r$ false-positive labels are sampled *uniformly without replacement* from the set of non-true classes and added to the candidate set, while the unique ground-truth label is always retained. Hence the candidate set has size exactly $r + 1$, contains the true label, and contains no duplicates among false positives. On the 5-class MNIST-MIPL and FMNIST-MIPL datasets, this construction is well-defined only for $r \leq 3$ – the cases $r = 4$ and $r = 5$ would saturate the candidate set, and the corresponding stress test is therefore reported on the 25-class SIVAL-MIPL dataset (Section 5.4).

**Key-instance evaluation metrics.** We formalize how the ability to "find the key instance" is quantified in Figure 1 and in the experiments below. Given a trained bag-level model with attention scores $\boldsymbol{\alpha}$ over the instances of a bag, let $\mathcal{K}_i \subseteq \{1, \ldots, n_i\}$ denote the *key-instance set* of bag $i$,

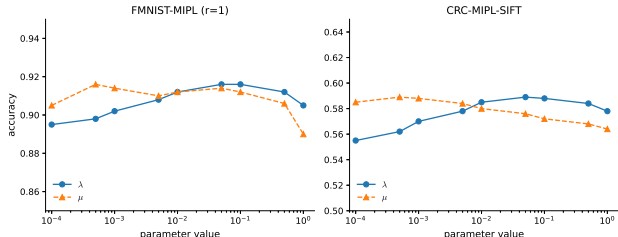

*Figure 3.* Performance of AGOPMIPL with varying $\mu$ and $\gamma$.

defined on the synthetic MNIST-style benchmarks as the set of instances that carry the true bag label.

- **Top-1 hit rate**: the indicator $\mathbb{I}(\arg\max_k \alpha_{i,k} \in \mathcal{K}_i)$ averaged over bags, i.e., the fraction of bags in which the highest-attention instance belongs to the key-instance set.
- **Average rank**: the mean rank, under descending attention, of the best-ranked element of $\mathcal{K}_i$, averaged over bags. Smaller is better.

When a bag contains multiple instances that are positive for the true label, all of them are placed in $\mathcal{K}_i$ rather than only one designated representative; Top-1 hit rate then asks whether the highest-attention instance falls into this set, and Average Rank uses the best rank achieved by any element of $\mathcal{K}_i$.

### 5.1. Results

AGOPMIPL achieves the highest or comparable accuracy in most benchmark settings in Table 2. It is best on all SIVAL-MIPL settings and on most MNIST-MIPL and FMNIST-MIPL settings, while FastMIPL remains competitive and obtains higher accuracy on Birdsong-MIPL at $r = 2, 3$ and on FMNIST-MIPL at $r = 3$. This pattern suggests that AGOP is most useful when candidate-label noise and key-instance localization are both difficult. Table 2 compares AGOPMIPL with five MIPL baselines: FastMIPL, ProMIPL, MIPLGP, DEMIPL, and ELIMIPL. The results of PLL and MIL algorithms are reported in the supplementary material.

Table 3 reports the results on real-world datasets. AGOPMIPL achieves the highest accuracy on all four datasets, with improvements of 9.6% on C-Row, 6.4% on C-SBN, 4.8% on C-SIFT, and 25.9% on C-KMeans. This improvement may be related to two factors: (i) the prior assumptions used by ProMIPL and FastMIPL are harder to satisfy on the more complex real-world feature distributions, leading to their degradation; (ii) AGOP-based feature learning adapts the embedding metric directly to the data and, combined with feature prototypes, identifies key instances and disambiguates candidate label sets more effectively under such

| Dataset | #bag | #ins | max #ins | min #ins | avg. #ins | #dim | #class | avg. #CLs |
|---|---|---|---|---|---|---|---|---|
| MNIST-MIPL | 500 | 20,664 | 48 | 35 | 41.33 | 784 | 5 | 2,3,4 |
| FMNIST-MIPL | 500 | 20,810 | 48 | 36 | 41.62 | 784 | 5 | 2,3,4 |
| Birdsong-MIPL | 1,300 | 48,425 | 76 | 25 | 37.25 | 38 | 13 | 2,3,4 |
| SIVAL-MIPL | 1,500 | 47,414 | 32 | 31 | 31.61 | 30 | 25 | 2,3,4 |
| CRC-MIPL-Row (C-Row) | 7,000 | 56,000 | 8 | 8 | 8 | 9 | 7 | 2.08 |
| CRC-MIPL-SBN (C-SBN) | 7,000 | 63,000 | 9 | 9 | 9 | 15 | 7 | 2.08 |
| CRC-MIPL-KMeansSeg (C-KMeans) | 7,000 | 30,178 | 6 | 3 | 4.31 | 6 | 7 | 2.08 |
| CRC-MIPL-SIFT (C-SIFT) | 7,000 | 175,000 | 25 | 25 | 25 | 128 | 7 | 2.08 |

*Table 1.* Experimental datasets with their characteristics

| Algorithm | $r$ | MNIST | FMNIST | Birdsong | SIVAL |
|---|---|---|---|---|---|
| FastMIPL | 1 | .999±.002 | .911±.022 | .790±.009 | .708±.030 |
|  | 2 | .998±.004 | .901±.027 | **.770±.010** | .630±.026 |
|  | 3 | .975±.074 | **.758±.071** | .767±.022 | .555±.031 |
| ProMIPL | 1 | .999±.003 | .921±.024 | .776±.015 | .682±.032 |
|  | 2 | .999±.003 | .889±.022 | .719±.018 | .633±.023 |
|  | 3 | .783±.116 | .659±.041 | .694±.021 | .539±.024 |
| ELIMIPL | 1 | .991±.005 | .904±.016 | .770±.019 | .676±.025 |
|  | 2 | .989±.013 | .843±.026 | .745±.017 | .615±.023 |
|  | 3 | .749±.148 | .701±.053 | .717±.019 | .599±.025 |
| DEMIPL | 1 | .977±.008 | .883±.019 | .741±.015 | .631±.042 |
|  | 2 | .944±.027 | .822±.026 | .702±.026 | .551±.056 |
|  | 3 | .711±.088 | .656±.027 | .694±.024 | .502±.017 |
| MIPLGP | 1 | .951±.019 | .846±.031 | .714±.026 | .669±.020 |
|  | 2 | .818±.033 | .792±.027 | .671±.015 | .614±.023 |
|  | 3 | .623±.062 | .669±.052 | .626±.015 | .570±.031 |
| AGOPMIPL | 1 | .999±.003 | **.922±.013** | **.806±.016** | **.759±.020** |
|  | 2 | **1.00±.000** | **.904±.042** | .769±.043 | **.734±.017** |
|  | 3 | **.986±.005** | .752±.013 | .742±.025 | **.705±.015** |

*Table 2.* Classification accuracy on MIPL benchmark datasets. Results are reported as mean accuracy ± standard deviation over repeated splits/folds. The best result in each column is **bolded**.

| Algorithm | C-Row | C-SBN | C-KMeans | C-SIFT |
|---|---|---|---|---|
| ProMIPL | .435±.009 | .516±.012 | .565±.013 | .562±.011 |
| FastMIPL | .487±.038 | .573±.031 | .573±.013 | .526±.029 |
| ELIMIPL | .434±.008 | .510±.008 | .545±.013 | .539±.010 |
| DEMIPL | .410±.011 | .484±.013 | .523±.012 | .531±.013 |
| MIPLGP | .435±.006 | .335±.008 | .331±.014 | – |
| AGOPMIPL | **.534±.010** | **.610±.007** | **.722±.010** | **.589±.011** |

*Table 3.* Classification accuracy on CRC-MIPL datasets. The best result in each column is **bolded**.

| Algorithm | Accuracy | Paired $t$-test ($p$-value) |
|---|---|---|
| AGOPMIPL | **.986±.005** | – |
| FastMIPL | .975±.074 | $p < 0.05$ |
| ProMIPL | .783±.116 | $p < 2 \times 10^{-4}$ |
| ELIMIPL | .749±.148 | $p < 10^{-3}$ |
| DEMIPL | .711±.088 | $p < 7 \times 10^{-9}$ |
| MIPLGP | .623±.062 | $p < 6 \times 10^{-6}$ |

*Table 4.* Paired $t$-test on classification accuracy over repeated splits on MNIST-MIPL ($r = 3$).

distributions.

| Method | MNIST | FMNIST | C-KMeans |
|---|---|---|---|
| AGOPMIPL | **1.00±.000** | **.752±.013** | **.722±.010** |
| AGOPMIPL\A | .891±.042 | .621±.031 | .618±.025 |
| AGOPMIPL\P | .985±.012 | .698±.018 | .695±.015 |

*Table 5.* Ablation study on the effectiveness of AGOP and progressive disambiguation ($r = 3$)

We further perform a paired $t$-test on classification accuracy over repeated splits on MNIST-MIPL ($r = 3$); as reported in Table 4, the improvement of AGOPMIPL over each baseline is statistically significant. Together with the overall accuracy results, where AGOPMIPL outperforms the baselines on most benchmark and real-world datasets, this supports its effectiveness for MIPL tasks.

### 5.2. Ablation Study

To assess the contribution of the proposed AGOP-based feature learning method and the progressive label disambiguation strategy, we perform ablation studies on these two components. Experiments are conducted on the MNIST-MIPL, FMNIST-MIPL, and CRC-KMeans with $r = 3$, and the AGOP feature learning component and the progressive label disambiguation component are removed, respectively. The variants without these components are denoted as AGOPMIPL\A and AGOPMIPL\P, respectively. The results are shown in Table 5. The accuracy of AGOPMIPL\A is markedly lower than that of the full model, indicating that the AGOP component contributes substantially to performance. Removing the progressive disambiguation strategy (AGOPMIPL\P) causes a smaller but still notable drop in accuracy.

### 5.3. Parameter Sensitivity Analysis

The two tunable loss weights $\mu$ and $\gamma$ in Eqn. (1) are evaluated on MNIST ($r = 1$) and CRC-MIPL-SIFT (Figure 3); accuracy varies smoothly with both, indicating they are easy to tune.

## 5.4. Robustness under Heavy Candidate-Set Noise

The main results of Table 2 use $r \in \{1, 2, 3\}$, which is the regime in which the candidate set is non-degenerate on every benchmark. To probe robustness in a more demanding setting, we additionally evaluate $r \in \{3, 4, 5\}$ on the 25-class SIVAL-MIPL dataset, where the candidate set remains far from saturated even at $r = 5$. Following the noise-injection protocol above, $r$ false-positive labels are sampled uniformly without replacement from the non-true classes for each bag. We compare AGOPMIPL with two competitive bag-level baselines (FastMIPL, ELIMIPL) under identical splits and report mean $\pm$ std over 3 folds.

| Algorithm | $r = 3$ | $r = 4$ | $r = 5$ |
| --- | --- | --- | --- |
| FastMIPL | .563±.009 | .531±.009 | .449±.043 |
| ELIMIPL | .453±.016 | .446±.028 | .409±.023 |
| AGOPMIPL | **.693±.014** | **.654±.014** | **.544±.035** |

*Table 6.* Test accuracy on SIVAL-MIPL under heavy candidate-set noise. AGOPMIPL retains a substantial margin over both attention-based baselines as $r$ grows.

As reported in Table 6, AGOPMIPL maintains higher accuracy than both baselines as $r$ grows: even at $r = 5$, where the candidate set contains five false positives in addition to the true label, AGOPMIPL outperforms FastMIPL by nearly 10 accuracy points. This is consistent with Proposition 4.4, which predicts that AGOP attenuates the contribution of bag-specific false-positive directions while preserving the cross-bag-coherent task-relevant subspace.

## 5.5. Additional Studies

AGOPMIPL is robust to its design choices. On the *prototype budget*, accuracy varies by less than 1.1 points across $n_p \in \{q/2, q, 2q\}$ on SIVAL-MIPL, so we use $n_p = q$ as the default (Appendix A). On the *bag representation*, aggregating raw instance features $\mathbf{H}$ outperforms aggregating AGOP-transformed features $\mathbf{H}'$, indicating that AGOP is best used as attention guidance rather than a direct replacement for the aggregated content (Appendix B).

On the *computational side*, because $\mathbf{J}_k(\boldsymbol{z})$ is read analytically from a linear classifier head, the AGOP update costs only $0.17\%$–$0.31\%$ of one full training round across datasets ranging from 250 to 4900 bags, scales mildly with the bag-level feature dimension $d'$, and remains compatible with transformer-based instance encoders (Appendix C).

## 6. Conclusion

We propose AGOPMIPL, which uses AGOP to rescale the feature space toward discriminative directions and identify key instances under candidate-set noise, complemented by a progressive label disambiguation strategy. The key insight is that AGOP reshapes the feature space using the directions to which the trained classifier is most sensitive: these are dominated by the true-label signal, which is reinforced across the training set, rather than by random false-positive labels, so in expectation true-label directions are amplified and noise directions are attenuated. This makes the learned feature metric more robust to noisy candidates without requiring an explicit denoising procedure. Extensive experiments show that AGOPMIPL outperforms existing MIPL methods on benchmarks and improves accuracy by $4.8\%$–$25.9\%$ on the real-world CRC-MIPL datasets. These results suggest that the cross-sample-aggregation perspective on feature learning may be a useful design principle for other weakly supervised settings.

## Acknowledgements

This work was supported by the Natural Science Foundation of Wuhan (2023010201020229) and the Fundamental Research Funds for the Central Universities (No. NJ2023032).

## Impact Statement

This paper studies weakly supervised learning under candidate-set noise. The proposed method may be applicable to settings with imperfect labels, such as crowdsourced image annotation or histopathology classification. There are many potential societal consequences of our work, none of which we feel must be specifically highlighted here.

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

## A. Prototype Budget Sensitivity

By default, each class is clustered into $n_p = q$ centroids, and the resulting $qn_p$ centroids are embedded and stacked to form the prototype set. This centroid budget $n_p$ is a practical convention rather than a theoretical requirement. To assess robustness to it, we vary $n_p$ over a wide range relative to the number of classes $q$ on SIVAL-MIPL ($q = 25$, $r = 3$ false-positive labels).

| Prototype budget | Accuracy |
|---|---|
| $n_p = 12 \approx q/2$ | .682±.014 |
| $n_p = 25 = q$ (default) | **.693±.014** |
| $n_p = 50 = 2q$ | .689±.017 |

*Table 7.* Sensitivity to the prototype budget on SIVAL-MIPL ($r = 3$ false-positive labels, 3-fold). Performance is stable across budgets, and we use $n_p = q$ as the default.

Performance varies by less than 1.1 accuracy points across this $4\times$ range, indicating that the model is not narrowly dependent on the prototype budget. We therefore use $n_p = q$ as the default in the rest of the experiments.

## B. Bag Representation Choice

Our default forms the bag representation as $z_i = \mathbf{X}_i^\top \alpha_i$, aggregating the original instance features with attention weights computed by the dual-branch attention module. Two natural alternatives are to aggregate the AGOP-transformed features $\mathbf{X}_i'$ instead, or to concatenate the two representations. We test all three on SIVAL-MIPL ($r = 3$, 3-fold) and on the real-world CRC-MIPL-KMeansSeg dataset.

| Bag representation | SIVAL ($r$=3) | CRC-KMeans |
|---|---|---|
| $z = \alpha\mathbf{H}$ (default) | **.693±.014** | **.722±.010** |
| $z = \alpha\mathbf{H}'$ | .671±.016 | .714±.005 |
| $z = [\alpha\mathbf{H}; \alpha\mathbf{H}']$ | .680±.010 | .710±.003 |

*Table 8.* Effect of the bag-representation choice. Replacing raw features with AGOP-transformed features in the aggregation hurts performance on SIVAL, suggesting that AGOP is better used as attention guidance than as a direct substitute for the bag feature; concatenation provides only a marginal change.

These results suggest that AGOP-transformed features are most useful as *attention guidance* – shaping which instances the bag-level pooling listens to – rather than as a direct replacement for the aggregated features themselves. Concatenation preserves some complementary information but does not exceed the simpler default in three-fold average.

## C. Computational Overhead and Scalability

In the implementation of AGOPMIPL, the AGOP matrix $\mathbf{G}_k$ is computed on the bag-level representation $z \in \mathbb{R}^{d'}$, and the practical overhead is therefore governed by $d'$, the number of training bags, and the AGOP update frequency, rather than by the number of raw instances or the input dimensionality $d$.

**Overhead vs. dataset size.** At fixed $d' = 128$, we profile the AGOP-update overhead on three representative datasets and contrast it with the full per-round training time. The AGOP update is performed analytically from the classifier head, so it does not require finite-difference passes or the storage of per-instance Jacobians (Section 4).

| Dataset | # bags | $d'$ | Best Acc | Round-1 train (s) | Round-1 AGOP (s) |
|---|---|---|---|---|---|
| FMNIST-MIPL ($r$=3) | 250 | 128 | .712 | 133.53 | 0.42 |
| SIVAL-MIPL ($r$=3) | 750 | 128 | .680 | 546.38 | 1.37 |
| CRC-KMeansSeg | 4900 | 128 | .706 | 1970.02 | 3.40 |

*Table 9.* AGOP-update overhead vs. dataset size at fixed $d' = 128$. Across datasets ranging from 250 to 4900 training bags, the AGOP-specific overhead remains within $0.17\%$–$0.31\%$ of one full training round.

The AGOP overhead remains within $0.17\%$–$0.31\%$ of one training round across all three datasets, so the AGOP step is not a practical bottleneck even on the largest benchmark.

**Overhead vs. feature dimension $d'$.** We further sweep $d' \in \{64, 128, 256, 512\}$ on CRC-KMeansSeg (fold 1) to verify that the overhead grows mildly in $d'$.

| $d'$ | Best Acc | Round-1 train (s) | Round-1 AGOP (s) | AGOP overhead | AGOP peak mem (MB) |
|---|---|---|---|---|---|
| 64 | .703 | 1972.23 | 3.42 | 0.17% | 17.69 |
| 128 | .706 | 1970.02 | 3.40 | 0.17% | 19.05 |
| 256 | .714 | 2021.05 | 3.61 | 0.18% | 22.63 |
| 512 | .705 | 2047.95 | 3.77 | 0.18% | 30.28 |

*Table 10.* Sensitivity to the bag-level feature dimension $d'$ on CRC-KMeansSeg. Increasing $d'$ from 64 to 512 keeps performance essentially flat, while AGOP update time and peak memory grow only mildly.

Increasing $d'$ from $64$ to $512$ keeps both accuracy and AGOP overhead essentially flat, with peak AGOP memory rising from $17.7\,\text{MB}$ to $30.3\,\text{MB}$.

**Feasibility on transformer-based encoders.** The AGOP framework is not specific to CNN/MLP encoders. To verify that the pipeline remains trainable when the per-instance feature extractor is replaced with a transformer, we substitute the default encoder by a lightweight transformer (2 layers, 4 attention heads) and continue to apply AGOP at the bag-level $d'$-dimensional projection.

| Encoder | Dataset | # bags | $d'$ | Best Acc | Round-1 AGOP (s) |
|---|---|---|---|---|---|
| Default (CNN/MLP) | SIVAL-MIPL ($r=3$) | 750 | 128 | **.680** | 1.36 |
| Lightweight Transformer | SIVAL-MIPL ($r=3$) | 750 | 128 | .651 | 7.60 |

*Table 11.* AGOP at the bag-level projection layer remains compatible with a transformer-based instance encoder. The transformer variant is trainable and reaches $0.651$ vs. $0.680$ for the default encoder; the AGOP overhead remains a tiny fraction of total training time.

The transformer variant is trainable end-to-end and reaches $0.651$ test accuracy versus $0.680$ for the default encoder; the AGOP overhead remains $\sim 0.2\%$ of total training time. We do not claim that the transformer encoder is preferable here – the SIVAL features are low-dimensional tabular vectors for which CNNs/MLPs are well-suited – but the result demonstrates that the AGOP step itself is not architecture-specific.

**Runtime and peak memory of AGOP itself.** On CRC-MIPL-KMeansSeg (fold 1), one AGOP update takes about $19.0\,\text{s}$ in round 1 and $3.7\,\text{s}$ in round 2 with $\text{bag}_{\text{repr}} = \text{raw}$, and about $18.3\,\text{s}$ and $3.8\,\text{s}$ with $\text{bag}_{\text{repr}} = \text{rfm}$. Because the update involves only the bag-level head and is performed intermittently, the peak GPU memory dedicated to AGOP is small (Table 10). We therefore conclude that the AGOP step is computationally inexpensive in practice across the regimes considered.

