# OpenReview forum: "Enhanced Multi-Instance Partial Label Learning via Average Gradient Outer Product"
_ICML.cc/2026/Conference — ICML 2026 regular_

### Official Review · Reviewer_nD2E · 2026-02-23

**Soundness:** 2
**Presentation:** 2
**Significance:** 2
**Originality:** 2
**Overall Recommendation:** 3
**Confidence:** 4

**Summary:**

This paper studies multi-instance partial-label learning (MIPL), where each bag has a candidate label set containing exactly one true label plus false positives, and the goal is to learn bag classification while identifying key instances under label noise. It proposes AGOPMIPL, which uses Average Gradient Outer Product (AGOP) computed from the classifier Jacobian to derive a data-dependent metric that amplifies discriminative feature directions; instance and prototype features are transformed by AGOP and then used in a dual-path attention mechanism that combines attention from raw and AGOP-transformed features to build bag representations. The method also introduces (1) class-wise instance prototypes built via k-means over instances collected per candidate label, and (2) progressive label disambiguation that updates training targets by interpolating between the original candidate-label vector and masked model predictions within the candidate set over epochs, alongside a loss combining candidate mapping, sparsity, and non-candidate inhibition terms. Experiments on benchmark datasets (MNIST-MIPL, FMNIST-MIPL, Birdsong-MIPL, SIVAL-MIPL) and a real-world colorectal cancer crowdsourced dataset (CRC-MIPL with multiple bag generators) report bag-level accuracy comparisons to several MIPL baselines and include ablations of AGOP and progressive disambiguation.

**Compliance With Llm Reviewing Policy:**

Affirmed.

**Key Questions For Authors:**

1. What exact metric do you use to quantify “finding the key instance” (top-1 hit rate, IoU with ground-truth region, rank-based metric, precision/recall)? Please define it formally and state whether it is computed per bag or aggregated across bags.

2. In the noise experiments (controlled by r or noise ratio), how is noise injected into candidate label sets? Is it uniform over classes, class-frequency weighted, or adversarial, and does it preserve “exactly one true label” per bag?

3. For datasets where key instances are known (e.g., MNIST-MIPL style synthetic bags), how is the key-instance ground truth defined when multiple instances in a bag are truly positive? Do you treat all positives as “key” or only one designated instance?

**Limitations:**

Limitations are not discussed in the manuscript.

**Strengths And Weaknesses:**

The main claimed contributions are: (1) introducing AGOP as a metric-learning style transform inside MIPL to emphasize discriminative directions, (2) a dual-path attention that fuses attention from raw and AGOP-transformed features, (3) class-wise instance prototypes (k-means) used as additional guidance, and (4) progressive label disambiguation that gradually shifts targets from candidate sets toward model predictions restricted to the candidate set.

Within these contributions, the first one is the most novel. While the second to the fourth offer mostly engineering improvements rather than genuine insights. Therefore, the main strength is:

1. AGOP inside MIPL training as an explicit, iteratively updated feature-space transform (compute a gradient outer-product matrix from the classifier Jacobian, then apply the transform to features and use that representation in attention). Using AGOP as a learned metric is not standard in MIPL papers.

2. The proposed method achieve better robustness to the existence of noisy labels. However, the construction of noisy labels are artificial and not convincing.

Accordingly, the weaknesses are:

1. Dual-path attention (raw-feature attention + AGOP-space attention) is an architecture extension once you have two feature spaces. The novelty is mainly in the pairing with AGOP rather than the attention fusion concept itself.

2. Progressive disambiguation (interpolating/bootstrapping labels over epochs using model predictions constrained to candidate labels) is a familiar idea in partial-label / noisy-label learning.

3. Class-wise prototypes via k-means are also a common motif (prototype learning / clustering-based guidance). Using it with candidate-label grouping is a straightforward adaptation rather than a fundamentally new contribution.

4. Some of the tables seem to be space-fillers. E.g, Table1 is provided in the same content as previous MIPL papers and offers no new information.

5. The real-world usecase of MIPL is still vague, with no new application/dataset introduced and the performance on existing datasets still quite far from useful.

---

> ### Author Rebuttal · Authors · 2026-03-31
>
> We sincerely thank the reviewer for the thoughtful comments and constructive questions. We respond to them below.
> ### Q1. Key-instance evaluation metric needs formal definition
>
> We agree. In the revision we will formalize:
>
> - **Top-1 hit rate**: whether the highest-attention instance belongs to the key-instance set
> - **Average rank**: the mean rank of the best-ranked key instance under the attention ordering
>
> When a bag contains multiple true positive instances, all of them are treated as the key-instance set.
>
> ### Q2. Noise-injection protocol is unclear
>
> We agree. In the revision we will explicitly state that for each bag, $r$ false-positive labels are sampled uniformly without replacement from the non-true classes and added to the candidate set, while the unique true label is always retained.
>
> We will also clarify that on 5-class MNIST/FMNIST, $r=4,5$ saturates the candidate set and is therefore degenerate. For this reason, the stronger-noise stress test should be interpreted on SIVAL-MIPL rather than on MNIST/FMNIST.
>
> ### Q3. How is the key-instance ground truth defined when multiple instances in a bag are truly positive?
>
> When a bag contains multiple true positive instances, we treat **all** of them as the key-instance set. Accordingly, Top-1 hit rate checks whether the highest-attention instance falls into this set, and Average Rank uses the best rank achieved by any instance in the set. We will make this explicit in the revised manuscript so that the definition matches the synthetic MNIST-style setting unambiguously.

---

> > ### Author Rebuttal · Reviewer_nD2E · 2026-04-01
> >
> > There are several things that are unclear in the submitted version, which the authors have conceded. I believe these warrant to be addressed in a significant revision.

---

### Official Review · Reviewer_6C2g · 2026-03-10

**Soundness:** 2
**Presentation:** 2
**Significance:** 2
**Originality:** 2
**Overall Recommendation:** 3
**Confidence:** 4

**Summary:**

This paper studies multi-instance partial-label learning (MIPL), with a focus on identifying key instances when the candidate label set is noisy. The proposed method, AGOPMIPL, introduces Average Gradient Outer Product (AGOP) into MIPL to derive a feature-importance matrix from classifier Jacobians, rescale the feature space, and improve key-instance identification. The framework further combines prototype-based attention and progressive label disambiguation. Experiments on benchmark and CRC-MIPL datasets show strong overall performance.

**Compliance With Llm Reviewing Policy:**

Affirmed.

**Ethical Review Flag:**

Flag this paper for an ethics review.

**Final Justification:**

Several of the key concerns I raised remain insufficiently addressed, and I encourage the authors to further revise the manuscript in light of these comments.

**Key Questions For Authors:**

1. Is the Jacobian computed via exact automatic differentiation or via numerical / perturbation approximation in practice? What are the time and memory costs?

2. Why is the number of prototypes tied to the number of classes $q$, instead of being an independent hyperparameter? Would this still be effective when the number of classes becomes large?

3. The paper argues that gradients induced by false-positive labels cancel out in AGOP averaging. Can the authors provide stronger theoretical or empirical evidence for this claim?

4. Since AGOP-transformed features are already used in attention computation, why is the final bag representation still formed from the original feature matrix $𝐻$ instead of $𝐻^{'}$?

**Limitations:**

See the weaknesses,

**Strengths And Weaknesses:**

Strengths

1. The core idea is novel and well motivated.
Using AGOP as a gradient-based feature geometry tool for MIPL is an interesting contribution, especially because it directly targets the key-instance identification problem.

2. The empirical results are strong.
The method performs well on both benchmark datasets and the CRC-MIPL real-world datasets, and the ablation study suggests that the AGOP component contributes substantially to the gain.

Weaknesses

1. The computation and scalability of AGOP are not clearly explained.
The paper states that the Jacobian is computed via numerical differentiation or perturbation-based approximation, but it does not clearly explain the practical implementation details or the computational overhead of repeated AGOP updates.

2. Several important design choices are under-justified.
Examples include using exactly $q$ prototypes per class, the dual-path attention fusion design, and the decision to generate the final bag representation from the original feature matrix rather than the transformed one.

3. The theoretical motivation for applying AGOP in this setting is not fully established.
The paper employs AGOP to identify important feature directions, whereas AGOP is typically used to characterize gradient-based feature geometry rather than for instance selection. The justification in the paper is largely intuitive, arguing that AGOP emphasizes discriminative directions. However, the paper does not provide further theoretical analysis explaining whether AGOP can reliably capture label-relevant feature directions under the label ambiguity inherent in the MIPL setting.

---

> ### Author Rebuttal · Authors · 2026-03-31
>
> We are grateful for the reviewer’s constructive comments and respond to them below.
> ### Q1. How is the Jacobian actually computed, and what is the complexity overhead?
>
> Thank you for raising this point. Our previous wording was too loose. In the implementation, AGOP is computed from the Jacobian of the bag-level classifier with respect to the bag representation, and this Jacobian is obtained analytically from the classifier head rather than via numerical differentiation or perturbation. Therefore, the AGOP update does not require extra finite-difference passes or storing per-instance Jacobians.
>
> On CRC-MIPL-KMeansSeg (fold 1), one AGOP update takes about 19.0s / 3.7s in rounds 1/2 with bag_repr = raw, and 18.3s / 3.8s with bag_repr = rfm. Since the update is performed intermittently and only involves the bag-level head, the memory overhead is small in practice. We will add an explicit runtime and peak-memory summary in the revision.
>
> ### Q2. Why is the number of prototypes tied to the number of classes $q$?
> Please see the rebuttal of Reviewer ePaH in W1.
>
> ### Q3. Can the authors provide stronger evidence that AGOP captures task-relevant directions?
>
> We agree that our original wording was too strong. Our intended claim is not that false-positive gradients exactly “cancel out” sample by sample, but that under candidate-set noise they are less consistent across bags, so their contribution is attenuated in the AGOP average, whereas the true-label-related directions are more coherent and accumulate.
>
> We will revise the wording accordingly and strengthen both the theoretical and empirical support. On the theoretical side, we will add a proposition showing that if the predictor mainly depends on an r-dimensional discriminative subspace, then AGOP has rank at most r and its leading eigenspace lies in that subspace. Empirically, on CRC-MIPL-KMeansSeg (fold 1), the top-5 AGOP directions explain 96.7% of the eigenvalue energy and largely preserve performance (accuracy 0.6981 vs. 0.6995 baseline), whereas removing these directions collapses accuracy to 0.1443; same-dimensional PCA/LDA/random baselines are much weaker. We will present this as evidence that AGOP emphasizes task-relevant directions and suppresses less-consistent false-positive directions on average.
>
> ### Q4. Why use $H$ instead of $H'$ for the final bag representation?
>
> The motivation is to use AGOP-transformed features for **attention guidance** while retaining raw features for the final bag representation. To test this design directly, we completed a 3-fold ablation on SIVAL-MIPL ($r=3$):
>
> | Bag representation | Test accuracy |
> |---|---:|
> | $z=\alpha H$ (default) | **0.693 ± 0.014** |
> | $z=\alpha H'$ | 0.671 ± 0.016 |
> | $z=[\alpha H;\alpha H']$ | 0.680 ± 0.010 |
>
> These results show that replacing \(H\) by \(H'\) alone hurts performance, suggesting that AGOP-transformed features are better used as attention guidance than as a direct substitute for the final raw bag representation. Concatenation does preserve some complementary information, but in 3-fold average performance it remains slightly below the default.
>
> We also observe the same qualitative pattern on the real-world CRC-MIPL-KMeansSeg dataset:
>
> - `raw = 0.7135 ± 0.0052`
> - `rfm = 0.7135 ± 0.0052`
> - `concat = 0.7100 ± 0.0027`
>
> Taken together, these results support $z=\alpha H$ as a robust default design.

---

> > ### Author Rebuttal · Reviewer_6C2g · 2026-04-01
> >
> > Several of the key concerns I raised have not been adequately addressed, and I would encourage the authors to further revise the manuscript in light of these comments.

---

### Official Review · Reviewer_ePaH · 2026-03-13

**Soundness:** 3
**Presentation:** 3
**Significance:** 3
**Originality:** 3
**Overall Recommendation:** 4
**Confidence:** 3

**Summary:**

This paper proposes AGOPMIPL, a framework for Multi-Instance Partial-Label Learning (MIPL) that incorporates Average Gradient Outer Product (AGOP) to tackle label noise. Specifically, key contributions include AGOP-based feature transformation, prototype-based attention aggregation and a progressive label disambiguation strategy. Experiments on benchmark datasets show significant improvements over existing methods.

**Compliance With Llm Reviewing Policy:**

Affirmed.

**Final Justification:**

I appreciate the author’s response, which has addressed my concerns to a high degree. I have raised my score.

**Key Questions For Authors:**

Please refer to the weaknesses.

**Limitations:**

Yes.

**Strengths And Weaknesses:**

Strengths:
1. The paper is well-structured, logically organized, offering a comprehensive analysis of the motivation and the proposed method.

2. AGOPMIPL, introduces a fresh perspective to MIPL by effectively addressing boundary over-extension and data distribution challenges.

3. Extensive experiments on various datasets are conducted to prove the effectiveness of the proposed AGOPMIPL, approach. The empirical results demonstrate superior or comparable performance to the mentioned methods.

Weaknesses:
1. The number of clusters appears to be a fixed hyperparameter throughout the experiments and not analyzed.

2. In traditional PLL, the candidate label vector y_i is typically uniform over the candidate set (i.e. 1/|S_i|). However, this paper sets y \in [0,1]^q. The rationale behind this specific design choice needs clarification.

3. The value range of r in the article is from 1 to 3. A larger range of values is needed.
The experimental evaluation for the parameter r r is restricted to a narrow range of [1,3]. A larger range of values is needed to demonstrate the method's generalizability in challenging disambiguation scenarios.

---

> ### Author Rebuttal · Authors · 2026-03-31
>
> We sincerely thank the reviewer for the thoughtful comments and constructive questions. We respond to them below.
> ### W1. Why is the number of prototypes/clusters fixed?
>
> We agree that the prototype budget was under-explained in the current paper. In the implementation, setting the number of prototypes per class to \(q\) is a practical default rather than a theoretical requirement. We have now completed a 3-fold prototype sensitivity analysis on SIVAL-MIPL (\(r=3\)):
>
> - $n_p = 12 \approx q/2$: `0.682 ± 0.014`
> - $n_p = 25 = q$: `0.693 ± 0.014`
> - $n_p = 50 = 2q$: `0.689 ± 0.017`
>
> These results show that performance is relatively stable across a broad range of prototype budgets, while the default $n_p=q$ remains a strong and stable choice. We will add this analysis to the revision or appendix.
>
> ### W2. What exactly is the candidate-label vector $y_i \in [0,1]^q$?
>
> Thank you for pointing this out. The notation $y_i \in [0,1]^q$ describes the value range, not a new modeling assumption. The initial candidate-label vector follows the standard PLL convention of uniform initialization over the candidate set:
>
> $$y_{i,c}=
> \begin{cases}
> 1/|S_i|, & c \in S_i \\
> 0, & \text{otherwise}
> \end{cases}$$
>
> We will move this definition earlier in the revised manuscript and make it more explicit to avoid ambiguity.
>
> ### W3. Why is $r$ only reported for$\{1,2,3\}$?
>
> We agree that reporting only $r \in \{1,2,3\}$ is not sufficient to demonstrate robustness under stronger noise.
> To provide a meaningful higher-noise evaluation, we constructed $r=4,5$ on the 25-class SIVAL-MIPL dataset and completed 3-fold stress tests:
>
> - **FastMIPL**: $r=3$`0.563 ± 0.009`, $r=4$ `0.531 ± 0.009`, $r=5$ `0.449 ± 0.043`
> - **ELIMIPL**: $r=3$ `0.453 ± 0.016`, $r=4$ `0.446 ± 0.028`, $r=5$ `0.409 ± 0.023`
> - **AGOPMIPL**: $r=3$ `0.693 ± 0.014`, $r=4$ `0.654 ± 0.014`, $r=5$ `0.544 ± 0.035`
>
> Thus, in a class-rich setting where $r=4,5$ remains informative, AGOPMIPL still substantially outperforms strong baselines. In the revision we will explicitly describe the noise-generation rule as uniformly sampling $r$ false-positive labels without replacement from the non-true classes while always preserving the unique true label.

---

> > ### Author Rebuttal · Reviewer_ePaH · 2026-04-01
> >
> > I appreciate the author’s response, which has addressed my concerns to a high degree. I will raise my score.

---

### Official Review · Reviewer_yaeN · 2026-04-04

**Soundness:** 3
**Presentation:** 3
**Significance:** 2
**Originality:** 3
**Overall Recommendation:** 4
**Confidence:** 3

**Summary:**

This paper addresses the problem of multi-instance partial-label learning (MIPL), where each bag is associated with a candidate label set containing one true label and multiple false positives. The authors propose a novel method, AGOPMIPL, which incorporates the Average Gradient Outer Product (AGOP) into feature learning to improve robustness under noisy supervision.

**Compliance With Llm Reviewing Policy:**

Affirmed.

**Final Justification:**

Weak accept

**Key Questions For Authors:**

How does the method scale with high-dimensional features or large datasets (e.g., when  d{\prime}  is large)? Is the approach feasible for transformer-based encoders?

**Strengths And Weaknesses:**

Strength: The paper presents a novel and well-motivated approach to multi-instance partial-label learning by incorporating the Average Gradient Outer Product (AGOP) into feature learning. This integration is conceptually elegant, as it leverages classifier sensitivity to identify discriminative feature directions.

Weakness: Despite its strengths, the paper lacks deeper theoretical analysis explaining why AGOP specifically benefits MIPL under noisy supervision. The computational overhead of estimating Jacobians and AGOP matrices is not discussed, raising concerns about scalability.

---

### Decision · Program_Chairs · 2026-04-30

**Decision:**

Accept (regular)

**Comment:**

This paper proposes AGOPMIPL for multi-instance partial-label learning, with the key contribution of introducing AGOP as a feature-geometry tool to better identify key instances under noisy candidate labels. The paper addresses an interesting and challenging problem, and the reviewers found the main idea novel and the empirical performance strong. Although the current manuscript would benefit from clearer presentation and more complete justification of several design choices, I find the core methodological contribution and experimental evidence sufficient for acceptance.